# Prebiopsy Magnetic Resonance Imaging Followed by Combination Biopsy for Prostate Cancer Diagnosis Is Associated with a Lower Risk of Biochemical Failure After Treatment Compared to Systematic Biopsy Alone

**DOI:** 10.3390/diagnostics15060698

**Published:** 2025-03-12

**Authors:** Shima Tayebi, Samuel Tremblay, Jason Koehler, Alon Lazarovich, Fernando Blank, Wei-Wen Hsu, Sadhna Verma, Abhinav Sidana

**Affiliations:** 1Department of Radiology, University of Cincinnati College of Medicine, Cincinnati, OH 45267, USA; 2Section of Urology, Department of Surgery, The University of Chicago Medicine & Biological Sciences, 5841 S. Maryland Ave, Rm P-217, Chicago, IL 60637, USA; 3College of Medicine, University of Cincinnati, Cincinnati, OH 45267, USA; 4Division of Urology, Department of Surgery, University of Cincinnati College of Medicine, Cincinnati, OH 45267, USA; 5Division of Biostatistics and Bioinformatics, University of Cincinnati College of Medicine, Cincinnati, OH 45267, USA

**Keywords:** prostate cancer, magnetic resonance imaging, oncological outcomes, biochemical failure

## Abstract

**Background:** Prostate cancer (PCa) diagnosis remains a complex field of study. Multiparametric magnetic resonance imaging (mpMRI) technology presents opportunities to enhance diagnostic precision. While recent advances in imaging and biopsy techniques show promise, the oncological implications of prebiopsy magnetic resonance imaging (MRI) and combination biopsy (ComBx) are not fully understood. This retrospective study evaluates the potential clinical impact of prebiopsy MRI and ComBx on PCa treatment outcomes. **Methods:** We conducted a comprehensive review of treatment-naïve patients undergoing prostate biopsy and subsequent radiation therapy (RT) or radical prostatectomy at the University of Cincinnati Health Center (2014–2020). Patients were categorized into two cohorts: those with prebiopsy mpMRI and ComBx versus those with systematic biopsy (SBx) alone. Patients with prostate-specific antigen (PSA) > 20 ng/mL were excluded. Biochemical recurrence (BCR) was defined as PSA ≥ 0.2 ng/mL post-prostatectomy or ≥2 ng/mL above nadir post-RT. **Results:** This study included 518 patients (189 SBx, 329 ComBx) with a median follow-up of 19.1 months. Median patient ages were 65.9 years (SBx) and 64.6 years (ComBx). The overall BCR rate was 10% with significantly lower rates in the ComBx group compared to SBx (6.4% vs. 16.4%, *p* < 0.001). Multivariable Cox regression analysis showed patients undergoing prebiopsy mpMRI with ComBx were 63% less likely to experience BCR (HR: 0.37, 95%CI 0.20–0.70, *p* = 0.002). **Conclusions:** Prebiopsy MRI followed by ComBx demonstrated lower BCR rates, suggesting improved PCa diagnosis and risk stratification. These findings highlight the potential of advanced imaging and biopsy techniques to benefit the management of PCa. Further longitudinal studies are needed to confirm the long-term clinical benefits of this approach.

## 1. Introduction

Prostate cancer (PCa) is a significant public health challenge worldwide. It is the second most commonly diagnosed cancer in men and the fifth leading cause of cancer-related death [1,2]. Globally, PCa contributes to a considerable disease burden, with approximately one in eight men being diagnosed during their lifetime [3]. Due to factors such as population growth and aging, the global burden of PCa is expected to rise substantially. These statistics underscore the urgent need for continued advancements in screening, early detection, and treatment strategies to mitigate the increasing burden of PCa worldwide [4].

Over the past decade, significant advancements have been achieved in PCa screening and diagnosis, particularly with the integration of advanced imaging that has improved detection accuracy [5,6]. These developments have transformed the diagnostic landscape, offering new possibilities for more precise and personalized approaches to PCa detection [7]. Traditionally, patients with elevated prostate-specific antigen (PSA) levels underwent non-targeted biopsies guided by transrectal ultrasound to confirm or exclude malignancy [8,9]. However, this method is limited by random sampling and is highly operator-dependent, potentially leading to underdiagnosis of clinically significant PCa (CsPCa) while risking overdiagnosis of clinically insignificant cancers, which may not require immediate therapeutic intervention [10,11,12,13]. Studies demonstrate that approximately 43% of prostatectomies are performed in patients later found to have indolent disease on histopathological analysis, while 60% of patients undergoing radical therapy (radiation or radical prostatectomy (RP)) present with grade group (GG) 1 cancers on preoperative biopsy [12,14]. These limitations of systematic biopsy (SBx) have necessitated the development and integration of complementary diagnostic modalities, most notably multiparametric magnetic resonance imaging (mpMRI) [13,15].

In recent years, prebiopsy mpMRI has emerged as a valuable tool that improves the detection of CsPCa while reducing the detection of clinically insignificant cases [16,17,18,19,20]. Utilizing mpMRI for men with elevated PSA represents a promising strategy that may simultaneously reduce unnecessary biopsies and enhance diagnostic precision. Beyond anatomical information, mpMRI provides critical data on tissue characteristics, including prostate volume, cellularity, and vascularity. Importantly, evidence suggests that mpMRI demonstrates preferential detection of higher-risk disease while systematically overlooking low-risk disease, making it particularly valuable as a prebiopsy assessment tool [17]. The superior soft tissue contrast and functional imaging capabilities of mpMRI allow for better visualization of suspicious lesions, enabling targeted sampling during biopsy procedures [21,22]. This approach represents a paradigm shift from systematic prostate sampling to targeted investigation of magnetic resonance imaging (MRI)-identified suspicious regions, which results in improved cancer detection rates and reduced procedural morbidity [16,17,18,23]. Therefore, prebiopsy mpMRI has become a key component of the diagnostic pathway and is now recommended in clinical guidelines [24].

Despite its ability to enhance diagnostic accuracy, the direct impact of prebiopsy MRI and targeted biopsy (TBx) on long-term oncological outcomes remains uncertain. While studies have demonstrated improved PCa detection rates and more accurate risk stratification with MRI-guided biopsies [25], the correlation between these advanced detection methodologies and patient outcomes requires further investigation. To address this gap, the present study aims to investigate the association between prebiopsy MRI-guided PCa diagnosis and the risk of biochemical recurrence (BCR).

## 2. Materials and Methods

### 2.1. Study Population

We conducted a retrospective review of treatment-naïve patients who underwent prostate biopsy followed by either radiation therapy (RT) or RP as their first treatment at the University of Cincinnati Health Center between 2014 and 2020. The dataset approved by the Institutional Review Board included comprehensive demographic and clinical data, such as PSA levels, prostate volume, digital rectal examination (DRE) findings, biopsy type, and BCR status. Patients were categorized into two cohorts: those who underwent prebiopsy prostate mpMRI followed by combination biopsy (ComBx), which included both MRI-TBx and SBx, and those who underwent SBx alone without mpMRI. Patients with PSA levels > 20 ng/mL were excluded from the study, as institutional protocols favored immediate SBx in such cases. Following the American Joint Committee on Cancer (AJCC) guidelines, we used the GG classification system for cancer grading. In our study, to ensure standardized assessment, CsPCa was defined as GG 2 or higher, a threshold selected based on the recommendations of the American Urological Association, which designates GG 2 and above as clinically significant [26,27,28,29].

### 2.2. MRI Details

mpMRI was conducted using a 1/5 or 3.0 T scanner (Achieva, Philips Healthcare, Amsterdam, The Netherlands). The imaging protocol comprised triplanar T2-weighted sequences, axial dynamic contrast-enhanced (DCE) sequences, axial diffusion-weighted imaging (DWI) with apparent diffusion coefficient (ADC) mapping, and, optionally, magnetic resonance spectroscopy (MRS). All MRI images were independently reviewed by two experienced radiologists, each with more than 10 years of expertise in prostate imaging. SBx was conducted following a sextant template to capture 12 random prostate biopsy cores guided by ultrasound transrectally by transrectal or transperineal approach. The ComBx approach involved both SBx and MRI-TBx (ARTEMIS™, Eigen Inc., Grass Valley, CA, USA).

### 2.3. Treatment and Follow-Up

Patients who underwent both diagnostic biopsy and subsequent treatment at our institution were included in this analysis. All radiation-based treatments (including external beam RT and brachytherapy) were categorized together as RT. Following the completion of their respective treatments, patients were scheduled for follow-up appointments every three months during the initial year and transitioned to biannual visits in subsequent years. Patients underwent a comprehensive assessment during each follow-up appointment, including PSA testing. To complement clinical assessments, patients underwent indicated imaging and/or biopsy procedures if they had experienced BCR to confirm the recurrence and guide subsequent management.

### 2.4. Outcomes

The primary outcome of this study was the assessment of BCR rate in patients following definitive treatment in two subgroups: patients who underwent prebiopsy mpMRI with ComBx and those who underwent SBx alone without mpMRI guidance. BCR was defined as a PSA level of ≥0.2 ng/mL following RP, or an increase of ≥2 ng/mL above the nadir following RT (Phoenix Criteria) [30].

### 2.5. Statistical Analysis

All statistical analyses were conducted using STATA Version 13.0 (StataCorp LP, College Station, TX, USA). Statistical significance was defined as a *p*-value of <0.05. Patient baseline characteristics were compared between the SBx and ComBx groups using the Mann–Whitney U test for continuous variables and the chi-square test for categorical variables. Kaplan–Meier analysis and log-rank tests were used to compare BCR between groups. Cox proportional hazards regression was conducted to identify predictors of BCR, adjusting for race, biopsy type, DRE findings, family history of PCa, biopsy results, PSA value, and age.

## 3. Results

In total, we analyzed a total of 518 men who presented with PSA levels of less than 20 ng/mL and underwent definitive PCa treatment. The median follow-up duration for this cohort was 19.1 months (IQR 6.1–37.8 months). Of these, 189 patients underwent pre-treatment SBx alone, while 329 received pre-treatment mpMRI followed by ComBx. Baseline characteristics are presented in Table 1. The median age of the entire cohort was 65.2 years (IQR, 59.7–70.0) with similar age distributions between the SBx group (65.9 years, IQR 60.5–71.0) and ComBx group (64.6 years, IQR 59.0–69.5). The majority of patients were Caucasian (64.1%) followed by African Americans (21.6%) and other races (14.3%). Additionally, DRE findings and family history of PCa were distributed evenly across both SBx and ComBx cohorts indicating no significant differences in baseline clinical characteristics between the groups. Regarding treatment modalities, similar proportions of patients underwent prostatectomy versus RT in both groups (SBx: 68.3% prostatectomy, 31.8% RT; ComBx: 66.3% prostatectomy, 33.7% RT; *p* = 0.642).

When comparing patients with and without BCR (Table 2), those who experienced BCR had higher median PSA values (9.0 vs. 6.8 ng/mL) and demonstrated a notably higher proportion of Group 5 disease (13.3% vs. 8.5%).

### Biochemical Recurrence Outcomes

The overall BCR rate was 10% (52 patients). Notably, the BCR rate was significantly lower in the ComBx group as only 6.4% (21 patients) experienced recurrence compared to 16.4% (31 patients) of the SBx group. This difference was statistically significant (*p* < 0.001) (Table 2). Kaplan–Meier survival analysis revealed significantly better BCR-free survival in patients who underwent prebiopsy MRI followed by ComBx compared to those who received SBx alone (*p* < 0.001) (Figure 1). The survival curves showed a clear divergence between the two groups as the ComBx group maintained consistently better outcomes throughout the follow-up period.

Multivariable Cox regression analysis confirmed ComBx as a significant predictor for BCR after PCa treatments (RP or RT) (HR:0.37 (95%CI 0.20–0.70); *p* = 0.002) even after controlling for race, DRE findings, family history of PCa, biopsy results, PSA value, and age (Table 3). Patients who had prebiopsy MRI followed by ComBx were found to be 63% less likely to experience BCR during the study follow-up period.

These findings further validate that patients who underwent prebiopsy MRI had a lower risk of biochemical failure. This suggests substantial long-term benefits of this diagnostic strategy.

## 4. Discussion

In this study, prebiopsy MRI followed by ComBx was associated with significantly lower rates of BCR compared to SBx alone with a marked reduction from 16.4% in the SBx group to 6.4% in the ComBx group. This finding remained consistent across different treatment modalities including both RP and RT patients. Importantly, the association between ComBx and reduced BCR rates persisted even after adjusting for other variables, such as race, DRE findings, family history, PSA value, and age. These findings highlight the broader implications of incorporating prebiopsy mpMRI into diagnostic and treatment-planning pathways. Not only does mpMRI improve the precision of PCa detection by targeting clinically significant lesions that may be missed by SBx alone, but it also facilitates better risk stratification. This allows for more personalized treatment planning, which potentially leads to improved long-term oncological outcomes.

The role of advanced imaging techniques, such as mpMRI, in both the diagnostic and prognostic pathways of PCa has gained increasing attention in recent years [18]. Several studies have explored the value of mpMRI in refining risk stratification, improving cancer detection, and predicting long-term oncological outcomes. Kasivisvanathan et al. conducted a landmark multicenter randomized trial that established the superiority of MRI-TBx over standard SBx for initial cancer detection [16]. Their study demonstrated that MRI-TBx detected more clinically significant cancers (38%) compared to SBx (26%) and reduced the detection of clinically insignificant disease. Similarly, Siddiqui et al. conducted a study from 2007 to 2014 on 1003 patients that evaluated the diagnostic accuracy of MRI-TBx compared to standard transrectal ultrasound-guided SBx in PCa detection, and they found increased detection of high-risk PCa with MRI-TBx [15]. While our study aligns with Kasivisvanathan and Siddiqui in recognizing the enhanced capabilities of prebiopsy MRI in detecting CsPCa, we extend the advantages of MRI-guided techniques to include improved long-term biochemical outcomes across both RP and RT treatment modalities.

Gandaglia et al. demonstrated the utility of predictive tools incorporating mpMRI and biopsy-derived parameters to identify patients at higher risk of early BCR after RP [31]. While their study highlighted the prognostic role of individual parameters such as seminal vesicle invasion and lesion diameter on mpMRI, our findings extend this understanding by showing that the diagnostic strategy itself—integrating prebiopsy MRI with ComBx—has a sustained impact on long-term biochemical outcomes across diverse treatment modalities. However, it is essential to note some differences between our studies. Gandaglia et al. focused specifically on patients undergoing RP and assessed the prognostic implications of MRI-TBx in predicting BCR [31]. In contrast, our study evaluated the association between prebiopsy MRI-guided PCa diagnosis and the risk of BCR following both RP and RT. The observed reduction in BCR rates among patients undergoing prebiopsy MRI and ComBx may be attributed to several factors. Firstly, prebiopsy MRI enables more accurate detection and localization of suspicious lesions within the prostate gland, which allows for TBx of suspicious areas identified on imaging. This targeted approach may improve the diagnostic yield and precision of the biopsy and facilitate more accurate risk stratification and subsequent treatment planning. Additionally, the integration of MRI findings with SBx enhances the overall sensitivity of PCa detection, which potentially reduces the likelihood of missing clinically significant tumors.

Other studies have investigated the relationship between prebiopsy MRI and PCa diagnosis, treatment selection, and pathological outcomes [32,33,34]. Bjurlin et al. found that using mpMRI for TBx reduces the sampling error associated with conventional biopsy by providing better disease localization and sampling [35]. This aligns with the findings of Li et al., who showed that patients who received prebiopsy MRI were more likely to undergo RP compared to other treatment modalities [32]. Although this treatment selection pattern could potentially explain the differences in BCR observed in our study, our subsequent analysis revealed that the benefit of prebiopsy MRI and ComBx persisted independent of treatment choice. Specifically, we found significant reductions in BCR rates for both patients undergoing RP and those receiving RT. However, it is important to consider that mpMRI does have limitations, especially in the areas of image quality and interpretation. Sosnowski discusses factors such as the impact of prostate spasms, patient obesity, and the presence of metal prostheses, which can degrade the quality of mpMRI studies and hinder accurate interpretation. Additionally, the complexity of the technology, such as the need for advanced MRI systems and highly experienced radiologists, also presents challenges that must be addressed to fully realize mpMRI’s diagnostic potential [36].

To the best of our knowledge, no prior studies have specifically investigated and reported on differences in BCR rates between the use of MRI versus no MRI in both RP and RT. Patel et al. found similar results with their cohort of 609 patients undergoing RP [37]. Their research demonstrated that the BCR rate was significantly improved in the MRI group after adjusting for age, PSA, RP GG, pT, pN, and positive surgical margin status (HR 0.64 (95%CI 0.41−0.99), *p* = 0.04). This suggests that the use of MRI in the diagnostic and treatment pathway can lead to better oncologic control after RP by allowing for more accurate patient selection and risk stratification. While Patel et al. focused exclusively on RP patients, our research expands on these insights by comparing BCR outcomes not only in RP patients but also in those undergoing RT. Improved outcomes in both treatment pathways reinforce the value of MRI in optimizing treatment planning and long-term oncologic outcomes for PCa patients in whichever treatment modality is chosen.

Our study has several important limitations. First, the evaluation of patient outcomes was conducted retrospectively, so it lacked randomization and potentially introduced selection bias. However, our large sample size and consistent findings across different subgroups help mitigate this concern to some degree. However, the absence of prospective randomization limits our ability to control for unknown confounding variables and may affect the generalizability of our results. Second, a notable proportion of patients had missing data, which may have influenced the outcomes. Third, this study encompasses a relatively small subset of patients and features only short-term follow-up. While this timeframe may not capture late recurrences, it does provide valuable insights into early biochemical outcomes, which are often predictive of longer-term results. Finally, while we demonstrate a correlation between ComBx and improved biochemical outcomes, we cannot definitively establish causation.

These limitations notwithstanding, our findings have significant implications for clinical practice and patient care. For policymakers and healthcare systems, our results support the current American Urological Association and European Association of Urology recommendations for mpMRI prior to biopsy [24,33]. For clinicians, our findings suggest that investing in prebiopsy MRI capabilities may lead to improved patient outcomes through better risk stratification and treatment planning. For patients, these results indicate that seeking care at centers offering prebiopsy MRI may lead to more favorable long-term outcomes. While some argue against the widespread use of prebiopsy MRI due to concerns about GG score differences, our study demonstrates oncological benefits independent of these considerations [38].

Our findings suggest that prebiopsy MRI followed by ComBx significantly improves biochemical outcomes in PCa patients. Future research should focus on longer-term follow-up studies to validate these findings and explore the specific mechanisms through which prebiopsy MRI leads to improved outcomes. Additionally, cost-effectiveness analyses comparing the increased upfront costs of MRI against potential savings from reduced recurrence rates would be valuable. Understanding these aspects will be crucial in optimizing the role of prebiopsy MRI in PCa care and potentially expanding its application to broader patient populations.

## 5. Conclusions

In summary, the use of prebiopsy MRI followed by ComBx for the detection of PCa showed a significant decrease in BCR rate after treatment compared to diagnosis by SBx alone. While ComBx appears to be associated with a reduced risk of BCR, it is possibly a reflection of the overall quality of care provided to these patients rather than a direct effect of the biopsy technique itself. The improved outcomes may be attributed to a combination of factors, including enhanced cancer detection precision, more accurate risk stratification, and optimized treatment planning capabilities afforded by advanced imaging. Our findings have important implications for clinical practice, such as supporting current professional guidelines recommending prebiopsy MRI and suggesting that the integration of advanced imaging into the diagnostic pathway may have benefits extending beyond initial cancer detection. Further research is warranted to explore the underlying factors contributing to this observed association and to determine the optimal diagnostic and treatment strategies for patients with PCa.

## Figures and Tables

**Figure 1 diagnostics-15-00698-f001:**
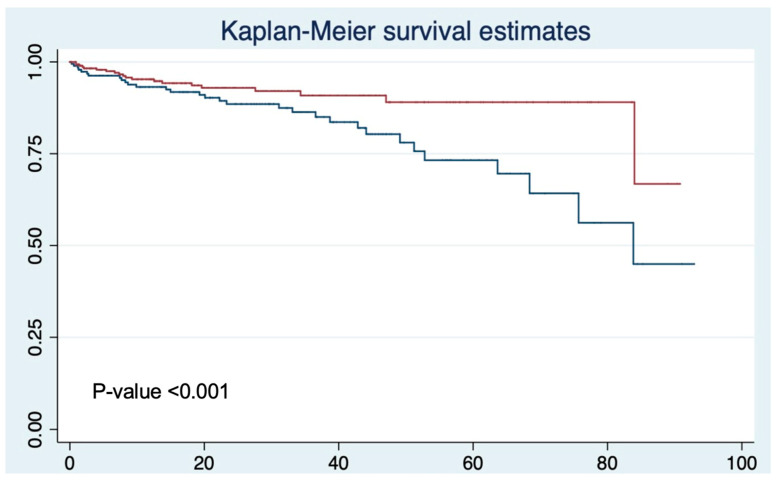
Kaplan–Meier curve demonstrating biochemical recurrence-free survival between prebiopsy MRI and combination biopsy group (red), and systematic biopsy group (blue, time in months).

**Table 1 diagnostics-15-00698-t001:** Baseline characteristics of patients.

	Systematic Biopsy (*n* = 189)	Combination Biopsy (*n* = 329)	*p*-Value
Median age year (IQR)	65.9 (60.5–71.0)	64.6 (59.0–69.5)	0.071
Race (%)			<0.001
White	91 (48.2)	241 (73.3)	
African American	89 (47.1)	81 (24.6)	
Positive family history of prostate cancer (%)	36 (19.2)	95 (29.7)	0.009
Median PSA ng/mL (IQR)	7.1 (5.2–10.6)	7.0 (5.3–102)	0.886
Median prostate volume mL (IQR)	38.2(30.0–46.0)	39.0 (30.0–51.0)	0.605
Abnormal DRE (%)	50 (27.0)	55 (17.0)	0.007
Grade group (%)			0.848
Group 1	16 (8.5)	20 (6.2)	
Group 2	91 (48.2)	165 (51.1)	
Group 3	41 (21.7)	65 (20.1)	
Group 4	20 (10.6)	38 (11.8)	
Group 5	21 (11.1)	35 (10.8)	
MRI highest PI-RAD score (%)			
PI-RAD 2		1 (0.3)	
PI-RAD 3		13 (4.4)	
PI-RAD 4		130 (44.1)	
PI-RAD 5		151 (51.2)	

IQR = Interquartile range; DRE = digital rectal exam; PI-RAD = prostate imaging-reporting and data system.

**Table 2 diagnostics-15-00698-t002:** Comparison of patient characteristics between those with and without biochemical recurrence.

	Biochemical Recurrence No (*n* = 457)	Biochemical Recurrence Yes (*n* = 52)	*p*-Value
Median age year (IQR)	65.2 (59.6–69.6)	65.9 (60.4–71.3)	0.369
Race (%)			0.459
White	300 (66.7)	32 (61.5)	
African American	150 (33.3)	20 (38.5)	
Positive family history of prostate cancer (%)	121 (26.5)	10 (19.6)	0.288
Median PSA ng/mL (IQR)	6.8 (5.2–9.9)	9.0 (5.8–13.6)	0.002
Median prostate volume ml (IQR)	39.0 (30.0–48.6)	37.9 (28.4–49.2)	0.607
Abnormal DRE (%)	97 (21.2)	8 (15.7)	0.354
Grade group (%)			<0.001
Group 1	35 (7.6)	1 (2)	
Group 2	242 (52.5)	14 (27.5)	
Group 3	92 (20)	14 (27.5)	
Group 4	53 (11.5)	5 (9.8)	
Group 5	39 (8.5)	17 (13.3)	
Biopsy type			<0.001
Systematic biopsy (%)	158 (33.9)	31 (59.6)	
Combination biopsy (%)	308 (93.6)	21 (40.4)	

PSA = prostate-specific antigen; DRE = digital rectal exam.

**Table 3 diagnostics-15-00698-t003:** Multivariable Cox regression analysis of factors associated with biochemical recurrence.

Variable Name	Hazard Ratio	95% Confidence Interval	*p*-Value
Race			
African American	0.98	0.50–1.92	0.949
Biopsy type			
Combination biopsy	0.37	0.20–0.70	0.002
Abnormal DRE	0.68	0.31–1.49	0.332
Biopsy grade group	1.61	1.25–2.07	<0.001
PSA value	1.09	1.02–1.17	0.008
Age	0.99	0.95–1.03	0.517
Positive family history	0.73	0.35–1.52	0.401

DRE = digital rectal exam; PSA = prostate-specific antigen.

## Data Availability

The data presented in this study are available on request from the corresponding author. The data are not publicly available due to privacy and ethical restrictions.

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
