# Peer review of "Prebiopsy Magnetic Resonance Imaging Followed by Combination Biopsy for Prostate Cancer Diagnosis Is Associated with a Lower Risk of Biochemical Failure After Treatment Compared to Systematic Biopsy Alone"

_diagnostics, 2025, doi:10.3390/diagnostics15060698_

Round 1

Reviewer 1 Report

Comments and Suggestions for Authors

Overall, this manuscript is well conceived and written. It suggests a significant benefit in terms of reduced biochemical recurrence (BCR) using a MRI-guided bioptic approach compared to systematic biopsy alone. Even if the study is timely and based on a large cohort, some points should be addressed. Firstly, the authors should better explain the criteria chosen for mpMRI. Also, they should provide a more detailed breakdown of both treatment decisions and Gleason Grade Group distribution within each biopsy group (SBx and ComBx). Moreover, the manuscript should provide more detail about the SBx and ComBx protocols. Additionally, the discussion could benefit from a more thorough comparison with literature data. In fact, the authors should better address both the studies that confirm the importance of a combined bioptic approach (doi: 10.5173/ceju.2016.734; https://doi.org/10.3390/diagnostics11010010) and those that show conflicting results or less clear benefits of MRI-targeted biopsy (doi: 10.5173/ceju.2016.e113; doi: 10.1007/s13244-015-0411-3; https://doi.org/10.21873/anticanres.15785). Moreover, the authors should provide some information about the MRI protocol, the experience of radiologists and the distribution of PI-RADS scores in the ComBx cohort. Furthermore, the authors should consistently use "Grade Group" (GG) instead of “Gleason group" throughout the text. Lastly, a spell and punctuation check should be performed.

Comments on the Quality of English Language

English Language needs moderate editing

Author Response

Dear Reviewer,

Thank you for your valuable feedback on our manuscript. We appreciate your thoughtful suggestions and have made the following revisions to address your points:

  1. "The authors should better explain the criteria chosen for mpMRI."

Thank you for pointing that out. As this is a retrospective study, there was no specific criteria for the MRI. The decision to use MRI was based on the surgeon's preference, the patient's preference, and insurance issues. As a result, some patients had MRI, while others did not.

  1. "Also, they should provide a more detailed breakdown of both treatment decisions and Gleason Grade Group distribution within each biopsy group (SBx and ComBx)."

Thank you for your comment. I have made the requested changes and now provide a detailed breakdown of both treatment decisions and Gleason Grade Group distribution within each biopsy group (SBx and ComBx). The Gleason Grade Group distribution is now included in Table 1, and the treatment decisions are clearly stated in the results section.

  1. "Moreover, the manuscript should provide more detail about the SBx and ComBx protocols."

I have added the necessary details about the SBx and ComBx protocols in Section 2.2, under the subheading "MRI and Procedure Details."

  1. "Additionally, the discussion could benefit from a more thorough comparison with literature data. In fact, the authors should better address both the studies that confirm the importance of a combined bioptic approach (doi: 10.5173/ceju.2016.734; https://doi.org/10.3390/diagnostics11010010) and those that show conflicting results or less clear benefits of MRI-targeted biopsy (doi: 10.5173/ceju.2016.e113; doi: 10.1007/s13244-015-0411-3; https://doi.org/10.21873/anticanres.15785)."

I appreciate your suggestion for a more thorough comparison with literature data. I have now addressed studies that confirm the importance of a combined bioptic approach as well as studies with conflicting results or less clear benefits of MRI-targeted biopsy. These references are now incorporated into the discussion.

  1. "Moreover, the authors should provide some information about the MRI protocol, the experience of radiologists, and the distribution of PI-RADS scores in the ComBx cohort."

The requested information about the MRI protocol, the experience of radiologists, and the distribution of PI-RADS scores in the ComBx cohort has been added to Section 2.2, under the "MRI and Procedure Details" subsection.

  1. "Furthermore, the authors should consistently use 'Grade Group' (GG) instead of 'Gleason group' throughout the text."

You are absolutely right, and I appreciate your attention to detail. I have now consistently used "Grade Group" (GG) instead of "Gleason group" throughout the manuscript.

  1. "Lastly, a spell and punctuation check should be performed."

I have revised the manuscript thoroughly to ensure proper spelling and punctuation.

Once again, thank you for your thoughtful and constructive feedback. I hope these revisions meet your expectations.

Regards,

Shima

Reviewer 2 Report

Comments and Suggestions for Authors

Thanks for the opportunity to review this manuscript.

The manuscript is well-structured and presents a comprehensive analysis of the impact of prebiopsy MRI followed by combination biopsy on prostate cancer treatment outcomes, with a particular emphasis on the reduction of BCR. The study provides a thorough overview, methodology, results, and discussion on the topic, concluding that the utilization of prebiopsy MRI followed by combination biopsy is advantageous in the management of prostate cancer patients. However, while the content is robust and the findings are significant, certain aspects require revision.

1. The introduction section should be a more detailed elucidation of how mpMRI and combination biopsy (ComBx) specifically address the limitations of systematic biopsy (SBx).

2. The definition of clinically significant prostate cancer (CsPCa) as GG 2 or higher is mentioned; however, there is no explanation of the grading system employed (e.g., Gleason score or ISUP), which may lead to confusion among readers unfamiliar with the terminology. 

Author Response

Dear Reviewer,

Thank you for your thoughtful review of our manuscript. We appreciate your positive comments and your constructive suggestions. Below are the revisions made in response to your feedback:

  1. "The introduction section should provide a more detailed explanation of how mpMRI and ComBx address the limitations of SBx."

Thank you for this suggestion. I have added a few sentences in the introduction to further explain how mpMRI and ComBx address the limitations of SBx, particularly in terms of improving detection accuracy and reducing sampling errors.

  1. "The definition of clinically significant prostate cancer (CsPCa) as GG 2 or higher is mentioned; however, there is no explanation of the grading system employed."

I have included an explanation of the grading system used (i.e., Gleason score and ISUP grading system) to clarify the definition of clinically significant prostate cancer (CsPCa) as Grade Group 2 or higher. This should help unfamiliar readers with the terminology.

Once again, thank you for your valuable feedback. I hope these revisions improve the clarity of the manuscript, and I look forward to your further comments.

Best regards,

Shima